# Physical Responses During Matches of International Female Football Players with Cerebral Palsy According to Their Sport Classes

**DOI:** 10.3390/sports13040094

**Published:** 2025-03-24

**Authors:** Oier Berasategui, Javier Yanci, Raúl Reina, Matías Henríquez, Aitor Iturricastillo, Ibai Errekagorri, Daniel Castillo

**Affiliations:** 1Faculty of Education and Sport, University of the Basque Country (UPV/EHU), 01007 Vitoria-Gasteiz, Spain; oierberasategui@gmail.com; 2Research Group in Physical Activity, Physical Exercise and Sport (AKTIBOki), Department of Physical Education and Sports, Faculty of Education and Sport, University of the Basque Country (UPV/EHU), 01007 Vitoria-Gasteiz, Spain; javier.yanci@ehu.eus (J.Y.); aitor.iturricastillo@ehu.eus (A.I.); 3Society, Sports and Physical Exercise Research Group (GIKAFIT), Physical Education and Sport Department, Faculty of Education and Sport, University of the Basque Country (UPV/EHU), 01007 Vitoria-Gasteiz, Spain; ibai.errekagorri@ehu.eus; 4Sports Research Centre, Miguel Hernández University, 03202 Elche, Spain; rreina@umh.es; 5Escuela de Kinesiología, Facultad de Salud, Universidad Santo Tomás, Santiago 8320000, Chile; matiashenriq@gmail.com; 6Valoración del Rendimiento Deportivo, Actividad Física y Salud, y Lesiones Deportivas (REDAFLED), Faculty of Education, University of Valladolid, 42005 Soria, Spain

**Keywords:** soccer, women, brain impairment, match load, Paralympics, para sport

## Abstract

**Background/Objectives:** International-level competition opportunities have recently been introduced for female footballers with cerebral palsy (CP), highlighting a gap in the research on their physical performance during matches. The objectives of this study were (I) to describe the physical responses during the 2022 Women’s World Cup of football players with cerebral palsy (CP) and (II) to analyze the differences in physical responses based on the players’ sport class (i.e., FT1, FT2, and FT3). **Methods**: Physical responses were recorded using global positioning devices (GPS) during four official international matches. **Results**: The results showed that FT2 players covered more explosive distances than FT1 players (*p* < 0.05; ES = −0.82), and FT2 and FT3 players achieved higher maximum velocities than FT1 players (*p* < 0.01; ES = −1.16 and *p* < 0.05; ES = −1.41, respectively). Furthermore, FT2 players performed more accelerations (*p* < 0.05; ES = −0.82 to −1.01) and decelerations (*p* < 0.01; ES = −1.00) in the mid–high intensity zones than FT1 players. **Conclusions**: While the greater impairment of FT1 players may have influenced their lower physical responses in competition compared to FT2 and FT3 players, the absence of differences between FT2 and FT3 classes is a novel aspect that requires further scientific investigation.

## 1. Introduction

One of the main missions of the International Paralympic Committee (IPC) is to promote social inclusion by providing opportunities for sports participation for individuals with disabilities, from amateur to elite levels [1]. Particularly, in the field of Para sports such as cerebral palsy (CP) football, the International Federation of Cerebral Palsy Football (IFCPF) has recently taken on the responsibility of promoting equality among para-athletes with the eligible impairments of spastic hypertonia, ataxia, and dyskinesia [2]. This commitment aims to ensure inclusivity regardless of gender, ethnic origin, faith, or culture, focusing on footballers with motor problems due to CP or related underlying health conditions [2]. CP football is a widely practiced team Para sport with both men’s and women’s divisions. However, almost all of its international tournaments and scientific research have focused on the male modality. Nevertheless, female CP football has been growing exponentially as a competitive sport in recent years across all levels [3]. Specifically, the growth of the modality has been a collaborative effort, marked by the inception of a national championships in 2017 and progressing towards the inaugural IFCPF World Cup 2022 in Salou, Spain.

From a scientific standpoint, to understand the sports discipline and adequately prepare female football players with CP for optimal performance, one of the primary areas of study should be the competition itself [4,5,6]. In the case of male CP football, competition has been analyzed from various perspectives, such as examining the physical responses of the players [7,8,9], their physiological responses [10], and even the technical and tactical actions through observational match analyses [11]. However, to date, no research has been addressing the competition element in female CP football, creating a significant gap in the scientific literature, especially regarding the differences in female players compared to their male counterparts concerning the size of the field of play (female: 40 × 27 m vs. male: 70 × 50 m), the number of players (female: five-a-side vs. male: seven-a-side) and the duration of an official match (female: two halves of 25 min plus 10 min of rest time vs. male: two halves of 30 min plus 15 min of rest time). In this sense, the external physical demands, as well as the fatigue and tactical behavior of both conventional football players and football players with CP, vary depending on the individual playing space and playing time [10,12]. In this regard, this study would be the first to analyze the physical responses of female football players with CP in competition, which could identify the specific demands of the game. This knowledge could be crucial for periodizing recovery strategies and adjusting weekly training loads in this population of para-athletes. Additionally, it could provide the classifiers with a better comprehension of the game for the classification process of female footballers with CP, who play with a different pitch size (smaller) and number of players (lower) in the line-up compared with their male counterparts.

On the other hand, due to the significant role of classification in Para sports [13], the study of physical responses during football matches based on sport class has also gained particular scientific relevance in recent years [9,14]. Particularly, in male CP football, it has been noted that the physical responses of players during competition vary depending on the sport class (i.e., at the beginning classified as FT5-FT8 and currently as FT1-FT3) [8,9]. Previous studies have indicated that male players in the FT2 and FT3 sport classes cover distances at higher intensities and achieve a higher maximum velocity compared to players in the FT1 class, who present a higher severity of the impairments [9]. Furthermore, players in FT3 also perform more accelerations and decelerations during competition compared to players in the FT1 and FT2 classes [7]. This scientific knowledge has contributed to modifying and enhancing the sport classification processes based on the scientific evidence [2,13,15]. Although in recent years, the scientific literature has focused on analyzing differences in physical responses during competition in male football players with CP [7,8,9], there is also a lack of knowledge regarding the physical responses of female footballers, considering the differences in the game compared with male CP football, but also whether female footballers’ performances differ based on the sport classes of the players.

Given the significance of advancing scientific knowledge on competition in female CP football to align with the objectives established by the IPC and the IFCPF [1,16], conducting research on this team Para sport modality is essential. Therefore, the main objectives of this study were (i) to describe the physical responses of female football players with CP during their first international-level competition and (ii) to analyze the differences in their physical responses based on the sport classes of the players (i.e., FT1, FT2, and FT3). Considering that differences in physical responses among male football players with CP based on the sport class have been previously observed [9], we hypothesized that differences in physical responses will also exist among female football players with CP, because of the level of impairment that is expected for each sport class.

## 2. Materials and Methods

### 2.1. Participants

Thirty-five international-level [17] female football players with CP (24.9 ± 8.3 years; 163.5 ± 7.0 cm; 59.0 ± 13.7 kg; 22.2 ± 3.5 kg·m^−2^) from top five national teams that competed in the International Federation of CP Football (IFCPF) 2022 World Cup (Salou, Spain) were included, representing nearly (77.78%) all international players who were active in this team Para sport when the study took place. All players trained 2–3 days per week and had an average of 4.76 ± 4.26 years. of enrollment in a national team program. The inclusion criteria for study participation included having a valid IFCPF license, being selected by their country/national team, competing internationally in the 2022 IFCPF World Cup (Salou, Spain), and regularly playing as a field player. The exclusion criteria were any type of injury in the three months prior to data collection or regular play as a goalkeeper, due to their specific role during the match. Participants were grouped based on their sport class into FT1 (n = 12; 19 individual observations; 27.9 ± 10.6 years.; 161.6 ± 7.4 cm; 54.0 ± 10.1 kg; 20.5 ± kg·m^−2^), FT2 (n = 19; 25 individual observations; 22.6 ± 6.0 years.; 164.8 ± 6.5 cm; 61.5 ± 14.8 kg; 22.5 ± 4.3 kg·m^−2^), and FT3 (n = 4; 4 individual observations; 25.5 ± 4.8 years.; 165.0 ± 7.9 cm; 67.6 ± 17.1 kg; 24.5 ± 4.0 kg·m^−2^). Participants were informed about the study procedures, could withdraw at any time, and were aware of the study risks before signing an institutionally approved informed consent document to participate. All procedures in this study were approved by the Ethics Committee of Miguel Hernández University (reference number DPS.RRV.03.17).

### 2.2. Procedures

Physical responses were recorded during official matches of the international competition (n = 35 players, 4 official matches, and 48 individual observations—i.e., physical responses monitored in a single game by one player—range = 1–3 matches per player). All variables were normalized based on the time played by each player in each match [9], except for the maximum velocity and maximum acceleration and deceleration variables (due to them representing absolute measures that are independent of playing time). All analyzed official matches were played five-a-side in the same sports facilities, using an artificial turf field of 40 × 27 m. Each match’s duration was 50 min of regular time (divided into two halves of 25 min) [18]. There was no extra time in any of the analyzed matches. The mean playing time for all players was 31.0 ± 15.4 min (range = 2.0–57.0 min).

### 2.3. Physical Variables

The physical responses of the football players were recorded during the matches with WIMU PRO devices (RealTrack Systems, Almería, Spain) using Global Positioning System (GPS) technology, operating at a sampling frequency of 10 Hz. This equipment and its measurements are valid and reliable using GPS for time–motion analyses in football [19] and were awarded with the FIFA Quality Performance certificate. It should be noted that these devices also had been used previously for measuring the physical responses of football players with CP [20]. Participants wore a vest, and the device was inserted on a vertical position between the players’ shoulder blades into a specially designed harness, placed before the warm-up for each match. Before the start of each match, players underwent a 20–25 min warm-up with their team, including running, progressive sprints, and stretching. These warm-up data were not included in the study analysis. Data from the half-time interval between the two halves of the match (i.e., 15 min) were also excluded from the analysis. The records were downloaded using the SPRO software (version 983, RealTrack Systems, Almería, Spain) after the end of each match. The total distance (TD) (m·min^−1^) [7,9], explosive distance (ED) (total distance traveled with an acceleration greater than 1.12 m·s^−2^) (m·min^−1^) [20], maximum velocity achieved (Vel_max_), distance (m·min^−1^) covered at different intensities [7] (walking: <6.0 km·h^−1^; jogging: 6.01–12.0 km·h^−1^; low-intensity running (LIR): 12.01–18.0 km·h^−1^; medium-intensity running (MIR): 18.01–21.0 km·h^−1^; high-intensity running (HIR): 21.01–24.0 km·h^−1^; and sprinting: 24.01–50.0 km·h^−1^), total number of accelerations (Acc) and decelerations (Dec) (n·min^−1^), maximum accelerations (Acc_max_) and decelerations (Dec_max_) reached (m·s^−2^) [7], and distances covered at different accelerations (Z1: 0/1 m·s^−2^; Z2: 1/2 m·s^−2^; Z3: 2/3 m·s^−2^; Z4: >3 m·s−^2^) and decelerations [7,9] (Z1: 0/−1 m·s^−2^; Z2: −1/−2 m·s^−2^; Z3: −2/−3 m·s^−2^; Z4: >−3 m·s^−2^) were calculated from the GPS-derived data. The player load (PL) was calculated as a vector magnitude representing the sum of accelerations recorded in the anteroposterior, mediolateral, and vertical planes [21].

### 2.4. Statistical Analysis

Results are presented as mean and standard deviation (SD). The Kolmogorov–Smirnov and Levene tests were applied to assess the data normality and homogeneity of variances, respectively. Parametric statistics were used for examined variables with a normal distribution of data and equal variances. However, non-parametric statistics were employed for MIR and HIR variables of distances covered at different speeds and zones 3 and 4 for accelerations and decelerations. To analyze differences between the FT1, FT2, and FT3 groups, a one-way analysis of variance (ANOVA) with Bonferroni post hoc comparison was used for variables with a normal distribution. For variables without a normal distribution, the non-parametric Kruskal–Wallis Test with Holm post hoc comparison was applied. The effect size (ES) was calculated using Cohen’s d [22], with values above 0.8, between 0.8 and 0.5, between 0.5 and 0.2, and below 0.2 considered large, moderate, small, and trivial, respectively [22]. Data were analyzed using the JASP software for Windows (version 0.16, University of Amsterdam, Amsterdam, The Netherlands). Statistical significance was set at *p* < 0.05.

## 3. Results

Table 1 presents the results for the total distance covered, explosive distance (total distance covered with an acceleration greater than 1.12 m·s^−2^), maximum velocity achieved, and distance covered at different intensities during matches for all the female players with CP, as well as differences based on their sport classes. The results indicate that FT2 players covered a greater explosive distance (*p* < 0.05; ES = −0.82; large) and achieved a higher maximum velocity than FT1 players (*p* < 0.01; ES = −1.16; large). Additionally, FT3 players reached a higher maximum velocity than FT1 players (*p* < 0.05; ES = −1.41; large). No significant differences were observed in any analyzed variable between the FT2 and FT3 sport classes.

Table 2 presents the results for total accelerations and decelerations and maximum accelerations and decelerations, categorized into the different intensity zones, as well as the PL for the entire group of players, along with differences between sport classes. The study results indicate that FT2 players perform more accelerations in Z3 (*p* < 0.05; ES = −0.82; large) and Z4 (*p* < 0.01; ES = −1.01; large), and decelerations in Z3 (*p* < 0.01; ES = −1.00; large) than FT1 players. No significant differences were observed for any of the variables between FT1 and FT3 or between FT2 and FT3 players.

## 4. Discussion

The main objective of this study was to compare the physical responses of female football players with CP based on their sport class (i.e., FT1, FT2, and FT3) during the first international-level competition held for this group at the 2022 IFCPF World Cup. Despite exhaustive analysis of physical responses in official matches in male CP football in recent years [7,8,9], this is the first study to examine the physical responses during competition in female football players with CP. The main findings of this study showed that FT2 and FT3 players achieved higher maximum velocities than FT1 players. Additionally, FT2 players covered more explosive distances than FT1 players. Similarly, FT2 players performed more accelerations in Z3 and Z4 and more decelerations in Z3 than FT1 players. However, no significant differences were found in physical responses between FT2 and FT3 players. Understanding the physical responses in competition and the differences between sport classes in female CP football would provide insights into the workloads of this sport modality and establish reference values for physical conditioning specialists to prepare female players for competition. Moreover, this study could also assist classification panels during the observation assessment processes during the game.

Our results do not align with a previous study conducted with male footballers with CP [9], which involved 42 players from different national teams during the 2016 World Championship. In that study, no significant differences were found between sport classes in maximum velocity or distances covered at different intensities. However, the results of this study partially coincide with those obtained in a study by Reina et al. [14], which showed significant differences in maximum velocities between the FT1 and FT2 classes and FT1 and FT3 players, walking distances between FT1 and FT2 players and FT1 and FT3 players, low-intensity distances between FT1 and FT3 players, and sprint distances between FT1 and FT2 players and FT1 and FT3 players. Similarly to this study, Reina et al. [14] found no significant differences in these variables between FT2 and FT3 players. These results also align with a previous study by Reina et al. [8], where significant differences were found in maximum velocities between FT1 and FT3 players and in sprint intensity distances (>18 km·h^−1^) between FT1 and FT3 players. In contrast to the present study, Reina et al. [14] found significant differences between the FT2 and FT3 classes in the maximum velocity achieved. A smaller field size and the smaller number of players could explain the difference in results compared to the results for men, as female football players with CP may adopt different tactical roles. In this regard, in the male format (seven-a-side, field dimensions of 70 × 50 m), the individual interaction space is 250 m^2^, whereas in the female CP football format (five-a-side, field dimensions of 40 × 27 m), the individual interaction space is 108 m^2^. The fact that the individual interaction space is significantly smaller could have influenced the ability of female players to reach their maximum physical potential in distances covered at medium-to-high intensities. The fact that FT1 players in this study obtained lower values for explosive distance or maximum velocity than FT2 and FT3 players can be partially explained by the greater physical impairment that FT1 players have, resulting in greater limitations in the range and coordination of lower limb movements [23,24]. These players may also have lower functional capacity due to their impairment [25], limiting their ability to achieve higher maximum velocities or cover more explosive distances in competition. However, the results of this study did not show significant differences between sport classes in the distances covered at medium and high speeds. Possibly, the absence of differences between sport classes in these variables is due to female footballers with CP rarely reaching these intensities in matches, but also, the smaller field of play may also explain these findings. The distances covered at moderate, high, and sprinting intensities by players from all classes were close to zero (total observations: MIR = 0.17 ± 0.32 m·min^−1^; HIR = 0.02 ± 0.07 m·min^−1^; and sprinting = 0.00 ± 0.00 m·min^−1^). These results contrast with those obtained for female players in conventional football, where they cover greater distances at medium or high intensities [26,27,28]. These results highlight that during competition, female football players with CP rarely reach speeds exceeding 18 km·h^−1^. More studies are needed to determine whether the reasons for not covering distances at these speeds are due to impairment, training levels, or other factors that are inherent to the competition format (i.e., the field size, individual interaction space, number of players during the game, or duration of an official game). However, considering that significant differences were observed between the sport classes in maximum velocity and explosive distance, the results of this study can be relevant, indicating that FT1 players have more activity limitations compared to FT2 and FT3 players in these variables. In addition, it could be considered that the defined zones may not be the most correct zones for female players with CP and that they may be used differently than the ones used for male players with CP considering the differences between both CP football modalities.

On the other hand, the abilities to accelerate and decelerate have been defined as important variables for sports performance [29] and have been studied in various research studies on male CP football to understand the neuromuscular demands of this Para sport [8,9]. Furthermore, it has been described that the specific demands of speed changes in competition, based on sport classes and the type of coordination problems (including levels of spastic hypertonia), have implications for the training of CP football teams [7,30]. In this regard, physical performance in CP football is determined by the player’s ability to perform short, fast, and specific actions (such as accelerating, decelerating, sprinting, or changing direction), as well as to withstand repetitions of these actions [20]. However, up to this point, no research has analyzed the physical capabilities of accelerations and decelerations that female football players with CP achieve in an international-level competition. Novelly, in the present study, FT2 class players performed more accelerations in Z3 and Z4 and more decelerations in Z3 than FT1 class players. However, no significant differences were observed in accelerations and decelerations at any intensity (Z1-Z4) between FT1 and FT3 players or between FT2 and FT3 players. The differences found in accelerations and decelerations between FT1 and FT2 players may be explained by the differences in the impact of the impairment [30] of players in both sport classes. However, the absence of differences between FT1 and FT3 players or between FT2 and FT3 does not align with the results obtained for male footballers with CP. In reports with male football players with CP [8,9,14], it was found that overall, FT3 players performed more high-intensity accelerations (>2.78 m·s^−2^) than those in FT1, with a moderate to large effect size (*p* < 0.05; ES range = 0.50–1.18). Furthermore, it has also been reported that FT3 players performed more high-intensity accelerations than FT2 players (*p* < 0.05; ES range = 0.73–0.86; moderate-to-large) [8,9]. Reina et al. [8] also found that FT3 players performed a greater number of moderate decelerations (−1.00 to −2.78 m·s^−2^) than FT1 (*p* < 0.05; ES = 0.73) and FT2 players (*p* < 0.05; ES = 0.69). Considering that the sport classification is determined by the players’ impairments, the absence of differences in accelerations and decelerations between FT1 and FT3 players and between FT2 and FT3 players is surprising. It also seems that female players with CP accelerate and decelerate at low intensities. Possibly, the lack of differences between these classes could be due to the different sample sizes in the various classes, especially in the FT3 group (n = 4 players). The fact that the participation of FT3 players is limited by the competition regulations (i.e., no more than one FT3 and at least one FT1 player in the field of play) [18] has not allowed for the inclusion of more players from this sport class, and this aspect may have influenced the obtained results. Another explanation could be the recent development of female CP football, where the level of training and competitive experiences is lower compared to male footballers with CP and where FT3 players are relevant players for tactics and strategy. Taking into account that the abilities to accelerate and decelerate are useful for assessing activity limitations in male footballers with neurological impairments [8], the understanding of the demands of short-duration, high-intensity actions (such as accelerations and decelerations) in their female counterparts would provide relevant information to understand the type of activity and demands in female CP football competition. A different trend is observed compared to male CP football, where there are more FT2 classes and a higher number of FT3 players, while FT1 remains the minority. The limited significant differences between FT2 and FT3 players could be attributed to their higher functional ability, combined with a smaller pitch size and the specific constraints of the modality (e.g., five-a-side game), which may limit variations in physical variables. What stands out in this population is that there are more women with higher levels of impairment competing, reflected in the large number of competitors in this sport class. This is noteworthy, because one might expect teams to seek more borderline athletes or those with greater functionality as the sport develops and becomes more competitive. However, this knowledge could contribute to designing specific and effective training programs for players, so more studies with larger sample sizes, especially from FT3 players, are needed to understand whether there are differences between the FT1 and FT2 classes compared to the FT3 class. Also, it will be interesting to conduct longitudinal studies to explore the performance improvements of female football players with CP with the development of the discipline and the inclusion of new national teams at international-level competitions.

Although the results of this research provide practical and novel information for both physical trainers and classification teams regarding the physical performance in competition of female footballers with CP, it is necessary to consider certain study limitations. First, the participant sample size was small due to data being obtained at the 2022 World Cup, but it is worth mentioning that an international-level group of the top five world class national teams was tested. Particularly, the limited representation of players in the FT3 category constitutes a significant practical constraint that could have affected the obtained results, although competition rules limit each team to only having one FT3 player in the field of play [18]. Therefore, although the analysis was carried out with four FT3 players, they represented a high percentage of the total number of players in that sport class playing female CP football when the study was completed. However, future research should aim to replicate the study with a larger sample as women’s participation in international championships continues to increase. On the other hand, the disparity in sample size between players of different sport classes (a practical limitation due to the different rules for the game for female and male CP football) could introduce potential biases in interpreting the results. Another limitation is the lack of analysis of physiological responses and a match observational analysis. Considering that there is currently a limitation imposed by the IFCPF competition regulations on the use of heart rate wristbands, a possible solution could be to record these variables during simulated matches or request the necessary permissions from the relevant governing body to collect these data during official competitions. These suggestions will allow for a more complete and thorough understanding of the specific competitive demands of female CP football, providing deeper insights into the competition. Furthermore, considering that this pioneering research addresses female CP football, there is a need for additional studies to delve into the association between performance in functional and match tests (i.e., balance, coordination, change in direction ability tests, etc.) and consider contextual factors such as the type of tournament (i.e., national or international) and environmental conditions (i.e., sea level or altitude).

## 5. Conclusions

The results of this study seem to indicate that the greater impairment of FT1 players may have influenced their lower competition responses compared to FT2 and FT3 players. However, despite the fact that the impairment of FT2 players is bigger than that of FT3 class players, the results obtained in this study found no significant differences in any of the competition-related physical response variables between FT2 and FT3 players. Therefore, it seems necessary to conduct more studies with larger sample sizes that analyze the physical responses of female football players with CP during matches, focusing specifically on differences between the FT2 and FT3 (i.e., minimal impairment) sport classes. On the other hand, a differentiating finding between conventional female football and male CP football was that female football players with CP did not reach speeds higher than 18 km·h^−1^, and the distance at medium and high speeds was very low.

## Figures and Tables

**Table 1 sports-13-00094-t001:** Physical response variables related to the total distance covered, maximum velocity achieved, and distance covered at different intensities during football matches by female players with cerebral palsy.

	Total Sample (48 Observations)				Pairwise Comparisons		
		FT1	FT2	FT3	FT1 vs. FT2 ES	FT1 vs. FT3 ES	FT2 vs. FT3 ES
TD (m·min^−1^)	66.96 ± 13.35	65.68 ± 18.04	67.04 ± 9.47	72.52 ± 8.42	−0.10	−0.51	−0.41
Vel_max_ (km·h^−1^)	17.10 ± 2.28	15.66 ± 1.79	17.97 ± 2.14	18.48 ± 1.97	−1.16 **	−1.41 *	−0.26
ED (m·min^−1^)	5.23 ± 1.88	4.35 ± 1.37	5.82 ± 2.02	5.79 ± 1.89	−0.82 *	−0.81	0.02
Distance intensities (m·min^−1^)							
Walking (<6 km·h^−1^)	38.53 ± 6.82	40.41 ± 9.86	37.30 ± 3.23	37.30 ± 4.91	0.46	0.46	0.00
Jogging (6.01–12 km·h^−1^)	23.18 ± 8.46	22.18 ± 9.98	23.15 ± 7.26	28.13 ± 7.93	−0.11	−0.70	−0.59
LIR (12.01–18 km·h^−1^)	7.14 ± 4.73	6.03 ± 5.34	5.03 ± 4.43	6.83 ± 2.78	−0.42	−0.17	0.26
MIR (18.01–21 km·h^−1^)	0.17 ± 0.32	0.05 ± 0.14	0.25 ± 0.40	0.25 ± 0.30	−0.63	−0.62	0.00
HIR (21.01–24 km·h^−1^)	0.02 ± 0.07	0.00 ± 0.00	0.03 ± 0.09	0.01 ± 0.03	−0.53	−0.23	0.30
Sprinting (24.01–50 km·h^−1^)	0.00 ± 0.00	0.00 ± 0.00	0.00 ± 0.00	0.00 ± 0.00	0.00	0.00	0.00

FT1 = sport class 1; FT2 = sport class 2; FT3 = sport class 3; ES = effect size; TD = total distance covered; Vel_max_ = maximum velocity achieved; ED = explosive distance; LIR = low-intensity running; MIR = moderate-intensity running; HIR = high-intensity running. Significant differences between sport classes: * *p* < 0.05, ** *p* < 0.01.

**Table 2 sports-13-00094-t002:** Accelerations and decelerations recorded for the entire group and differences between sport classes during official matches by female players with cerebral palsy.

	Total Sample (48 Observations)				Pairwise Comparisons		
		FT1	FT2	FT3	FT1 vs. FT2 ES	FT1 vs. FT3 ES	FT2 vs. FT3 ES
Acc (n·min^−1^)	33.39 ± 5.46	35.06 ± 7.69	32.78 ± 2.87	29.21 ± 1.37	0.43	1.10	0.67
Dec (n·min^−1^)	23.72 ± 7.16	26.75 ± 5.96	20.72 ± 7.41	22.05 ± 8.08	0.89	0.70	−0.20
Acc_max_ (m·s^−2^)	4.69 ± 1.84	4.60 ± 1.80	4.78 ± 1.94	4.60 ± 1.76	−0.90	0.01	0.10
Dec_max_ (m·s^−2^)	−5.55 ± 2.34	−5.52 ± 2.24	−5.70 ± 2.49	−4.75 ± 2.28	0.08	−0.33	−0.40
PL (AU·min^−1^)	0.76 ± 0.50	0.63 ± 0.48	0.82 ± 0.53	1.03 ± 0.32	−0.38	−0.81	−0.42
Acc/Dec Zones (n·min^−1^)							
Acc Z1 (0/1 m·s^−2^)	17.46 ± 3.60	18.25 ± 5.19	16.89 ± 1.89	17.27 ± 2.43	0.38	0.27	−0.10
Acc Z2 (1/2 m·s^−2^)	12.58 ± 3.88	12.60 ± 4.22	12.64 ± 3.94	12.08 ± 2.08	−0.01	0.13	0.14
Acc Z3 (2/3 m·s^−2^)	4.41 ± 2.08	3.40 ± 1.39	4.98 ± 2.05	5.60 ± 3.37	−0.82 *	−1.14	−0.30
Acc Z4 (>3 m·s^−2^)	0.61 ± 0.81	0.15 ± 0.25	0.89 ± 0.95	1.07 ± 0.67	−1.01 **	−1.26	−0.25
Dec Z1 (0/−1 m·s^−2^)	17.52 ± 3.67	18.12 ± 5.22	16.79 ± 1.97	19.27 ± 2.33	0.37	−0.32	−0.60
Dec Z2 (−1/−2 m·s^−2^)	10.81 ± 3.05	10.86 ± 3.35	10.45 ± 2.60	12.87 ± 3.81	0.13	−0.66	−0.80
Dec Z3 (−2/−3 m·s^−2^)	3.72 ± 1.64	2.86 ± 1.20	4.37 ± 1.75	3.81 ± 0.87	−1.00 **	−0.60	0.37
Dec Z4 (>−3 m·s^−2^)	0.90 ± 0.94	0.54 ± 0.61	1.15 ± 1.07	1.07 ± 1.10	−0.66	−0.58	0.08

FT1 = sport class 1; FT2 = sport class 2; FT3 = sport class 3; ES = effect size; Acc = total accelerations; Dec = total decelerations; Acc_max_ = maximum acceleration; Dec_max_ = maximum deceleration; Acc/Dec Zones = zones of accelerations and decelerations at different intensities; Acc Z1 = accelerations between 0 and 1 m·s^−2^; Acc Z2 = accelerations between 1 and 2 m·s−^2^; Acc Z3 = accelerations between 2 and 3 m·s^−2^; Acc Z4 = accelerations > 3 m·s^−2^; Dec Z1 = decelerations between 0 and −1 m·s^−2^; Dec Z2 = decelerations between −1 and −2 m·s^−2^; Dec Z3 = decelerations between −2 and −3 m·s^−2^; Dec Z4 = decelerations < −3 m·s^−2^; PL = player load. Significant differences between sport classes: * *p* < 0.05, ** *p* < 0.01.

## Data Availability

The data files that support the findings of this study are available from the corresponding author (DC) upon reasonable request.

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
