# Peer review of "Physical Responses During Matches of International Female Football Players with Cerebral Palsy According to Their Sport Classes"

_sports, 2025, doi:10.3390/sports13040094_

Round 1
Reviewer 1 Report
Comments and Suggestions for Authors
Thank you for the opportunity to evaluate this scientific article. I will briefly indicate for each chapter, my suggestions as follows:
In the Introduction:
A clarification is needed of the fact/reasons referring to previous studies that did not sufficiently cover the subject and how your research fills this gap.
In the Materials and Methods
It would be useful to indicate more details about the selection of participants in the sense that you need to clarify whether all players were at similar competitive levels to ensure comparability.
In the Results
Provide effect sizes consistently: Some sections discuss p-values ​​without mentioning effect sizes. Include them where relevant to provide a more complete interpretation.
Discussion
At the end of this chapter, explicitly and concisely state the limitations of this study
Conclusions
Some points are repeated from the Discussion, redundancy is not indicated in this case.
Author Response
- Thank you for the opportunity to evaluate this scientific article. I will briefly indicate for each chapter, my suggestions as follows:
Authors’ response (AR): We would like to thank the reviewer for the helpful advice and suggestions regarding this manuscript. In addition, we have found your criticism and recommendations very constructive. Also, we appreciate the work done in the revision of this manuscript, answering your concerns on a point-by-point basis. The changes to the paper have been made in red type in the new version of the manuscript.
- In the Introduction:
A clarification is needed of the fact/reasons referring to previous studies that did not sufficiently cover the subject and how your research fills this gap.
AR: Thanks for this suggestion. We have added information by rewriting the following sentences, to clarify that this study addresses the topic of the physical responses of female CP football for the first time, as previous studies only focused on the men's modality (Lines 65-72):
“In this regard, this study would be the first to analyze the physical responses of female CP football players in competition, which could identify the specific demands of the game. This knowledge could be crucial to periodize recovery strategies and to adjust weekly training loads in this population of para-athletes. Additionally, it could provide the classifiers a better comprehension of the game for the classification process of female footballers with CP, which play with a different pitch size (smaller) and number of players (lower) in the line-up compared with their male counterparts.
- In the Materials and Methods
It would be useful to indicate more details about the selection of participants in the sense that you need to clarify whether all players were at similar competitive levels to ensure comparability.
AR: Thank you for this comment. To expand the description of the players’ characteristics, it is mentioned that all players are international-level players and competed in the 2022 IFCPF World Cup. In addition, it has been added that the study sample represents 77.78% of the players in the biggest international female CP football tournament. As such, this sample ensures the representation of international female CP football players, and we can highlight the power of the results obtained in the study. To improve this section, we have included the following information (Lines 101-105):
“Thirty-five international-level [17] female CP football players (24.9 ± 8.3 yrs; 163.5 ± 7.0 cm; 59.0 ± 13.7 kg; 22.2 ± 3.5 kg·m-2) from top five national teams that competed in the International Federation of CP Football (IFCPF) 2022 World Cup (Salou, Spain), representing nearly (77.78%) all international players currently active in this team Para sport.”
- In the Results
Provide effect sizes consistently: Some sections discuss p-values ​​without mentioning effect sizes. Include them where relevant to provide a more complete interpretation.
AR: Thank you for this advice. The effect sizes (ES) were calculated using Cohen's d. The abbreviation ES is used in the Tables and the text of the manuscript. The Tables directly indicate the ES values, adding a * when p < 0.05 and ** when p < 0.01. In the text, the ES is included in the results marked with its interpretation, including their respective pvalue as suggested by the reviewer. Please see the results section.
- Discussion
At the end of this chapter, explicitly and concisely state the limitations of this study.
Conclusions
Some points are repeated from the Discussion, redundancy is not indicated in this case.
AR: Thanks for this recommendation. We have modified the limitations paragraph and the repeated points from the discussion in the conclusions to improve readability and the specific considerations of the study. We have included the following piece of information (Lines 321-328 and 332-336).
“Particularly, the limited representation of players in the FT3 category constitutes a significant practical constraint that could have affected the obtained results, although competition rules limit each team to only having one FT3 player in the field of play [18]. Therefore, although the analysis is carried out with four FT3 players, they represent a high percentage of the total number of players in that sport class playing female CP football when the study was done. However, future research should aim to replicate the study with a larger sample as women's participation in international championships continues to increase.”
“Considering that there is currently a limitation imposed by the IFCPF competition regulations on the use of heart rate wristbands, a possible solution could be to record these variables during simulated matches or request the necessary permissions from the relevant governing body to collect this data during official competitions.”
Reviewer 2 Report
Comments and Suggestions for Authors
Overall, I believe the manuscript is well organized and presents valuable information on an underserved population. This manuscript has the potential to add positively to this field of research. The comments/observations below are meant to aid in the refinement of the end product.
Abstract :
Edit: Internation-level competition opportunities has been recently introduced for female footballers with cerebral palsy (CP), existing a gap about their match physical performance.
To: International-level competition opportunities have recently been introduced for female footballers with cerebral palsy (CP), highlighting a gap in research on their match physical performance.
Statistics:
While Cohen’s d is used appropriately, a clearer discussion on how these findings translate into coaching and classification decisions would enhance applicability.
Tables and Figures:
Please consider adding bar charts or line graphs to enhance comprehension.
Discussion:
Consider elaborating on the potential reasons why female CP players exhibit different physical responses than male counterparts beyond field size and game format.
Limitations and Future research:
Well done acknowledging sample size limitations, particularly regarding FT3 players. However, it would be beneficial to discuss possible strategies for future research to mitigate these issues.
You might suggest exploring the inclusion of additional physiological metrics (e.g., heart rate variability, lactate threshold) to complement GPS data.
My overall perspective of the manuscript is that it presents significant findings and is a strong contribution to the field of para-sports research. With minor revisions it will be improved and considered for the next step in the publication process.
Author Response
- Overall, I believe the manuscript is well organized and presents valuable information on an underserved population. This manuscript has the potential to add positively to this field of research. The comments/observations below are meant to aid in the refinement of the end product.
Authors’ response (AR): We would like to thank the reviewer for the helpful advice and suggestions regarding this manuscript. In addition, we have found your criticism and recommendations very constructive. Also, we appreciate the work done in the revision of this manuscript, answering your concerns on a point-by-point basis. The changes to the paper have been made in red type in the new version of the manuscript.
- Abstract:
Edit: Internation-level competition opportunities has been recently introduced for female footballers with cerebral palsy (CP), existing a gap about their match physical performance.
To: International-level competition opportunities have recently been introduced for female footballers with cerebral palsy (CP), highlighting a gap in research on their match physical performance.
AR: Thank you for this proposal. We have substituted the initial sentence according to the reviewer´s suggestion (Lines 19-21):
“International-level competition opportunities have recently been introduced for female footballers with cerebral palsy (CP), highlighting a gap in research on their match physical performance”.
- Statistics:
While Cohen’s d is used appropriately, a clearer discussion on how these findings translate into coaching and classification decisions would enhance applicability.
AR: Thank you for this advice. Cohen's d is used to know how the magnitude differences between the independent sample means (i.e., FT1, FT2, and FT3) of our study are. The qualitative interpretation of the effect size is as follows: values above 0.8, between 0.8 and 0.5, between 0.5 and 0.2, and below 0.2 are considered large, moderate, small, and trivial, respectively. Classifiers must take into consideration those relevant variables that indicate moderate to large effect sizes which could indicate practical differences at the moment to differentiate between sport classes. However, this must be considered with caution due to the study limitations reported. To expand this information, authors have included the following piece of information (Lines 321-328):
“Particularly, the limited representation of players in the FT3 category constitutes a significant practical constraint that could have affected the obtained results, although competition rules limit each team to only having one FT3 player in the field of play [18]. Therefore, even though the analysis is carried out with four FT3 players, they represent a high percentage of the total number of players on that sport class playing female CP football. However, future research should aim to replicate the study with a larger sample as women's participation in international championships continues to increase.”
- Tables and Figures:
Please consider adding bar charts or line graphs to enhance comprehension.
AR: Thank you for this recommendation. We understand the reviewer’s suggestion. In this sense, we are willing to illustrate in figures the necessary results, although we consider that it would not be good practice to repeat information in text/tables and figures, and therefore, we are open mind to concrete suggestions to proceed with the design of the figures. Since this is the first study to describe the physical responses of female para-athletes with CP during competition matches, the authors believe that presenting performance data in Tables enhances clarity for readers. Tables provide a more illustrative format and facilitate a detailed presentation of statistical analyses, including effect size comparisons, thereby improving data interpretation.
- Discussion:
Consider elaborating on the potential reasons why female CP players exhibit different physical responses than male counterparts beyond field size and game format.
AR: Thank you for the comment. We have added the following reason comparing the male and female modalities and its participant physical responses (Lines 227-234):
“A smaller field size and the smaller number of players could explain the difference in results compared to the results in men, as female CP football players may adopt different tactical roles. In this regard, in the male format (7-a-side, field dimensions: 70×50 m), the individual interaction space is 250 m², whereas in the female CP football format (5-a-side, field dimensions: 40×27 m), the individual interaction space is 108 m². The fact that the individual interaction space is significantly smaller could have influenced the ability of female players to reach their maximum physical potential in distances covered at medium-to-high intensities”.
- Limitations and Future research:
Well done acknowledging sample size limitations, particularly regarding FT3 players. However, it would be beneficial to discuss possible strategies for future research to mitigate these issues.
AR: Thank you for this suggestion. We have added the following sentence (Lines 321-328):
“Particularly, the limited representation of players in the FT3 category constitutes a significant practical constraint that could have affected the obtained results, although competition rules limit each team to only having one FT3 player in the field of play [18]. Therefore, although the analysis is carried out with four FT3 players, they represent a high percentage of the total number of players in that sport class playing female CP football. However, future research should aim to replicate the study with a larger sample as women's participation in international championships continues to increase.”
- You might suggest exploring the inclusion of additional physiological metrics (e.g., heart rate variability, lactate threshold) to complement GPS data.
AR: Thank you for this suggestion. The reviewer is correct, the authors try to suggest the use of that kind of physiological metrics to complement the time motion data provided by GPS. However, according to the IFCPF rules of the game (https://www.ifcpf.com/tournaments/rules) it is not possible to include wristbands in official competitions. We have modified the sentence as follows (Lines 332-336):
“Considering that there is currently a limitation imposed by the IFCPF competition regulations on the use of heart rate wristbands, a possible solution could be to record these variables during simulated matches or request the necessary permissions from the relevant governing body to collect this data during official competitions.”
- My overall perspective of the manuscript is that it presents significant findings and is a strong contribution to the field of para-sports research. With minor revisions it will be improved and considered for the next step in the publication process.
AR: We appreciate the associate reviewers’ comments and propositions. We ensure that this study has been improved with this review, and we wish it progress for final acceptation and publication in Sports journal.
Reviewer 3 Report
Comments and Suggestions for Authors
Introduction
Lines 37-41. The sentence is too long and difficult to follow. Suggest breaking it into two sentences for clarity.
Lines 41-46. The transition from discussing IPC’s mission to the focus on male CP football feels abrupt. Consider adding a bridging sentence explaining why this disparity exists.
Lines 43-44. The sentence is grammatically incorrect. “In the game. recent years” seems to be a typo. Suggest rewriting for clarity: “Nevertheless, female CP football has been growing exponentially as a competitive sport in recent years across all levels.”
Lines 55-58. Good mention of differences between male and female CP football, but the connection to research gaps is weak. How do these differences impact performance, fatigue, or strategy? Add a sentence explaining why these factors warrant further study.
Lines 80-82. This sentence is poorly structured and difficult to understand.
Lines 82-85. The objectives should explicitly state how they address the identified research gaps.
Lines 86-89. The justification for this hypothesis is weak. Why is this expected? Are there physiological reasons or findings from male CP football that support this assumption? Consider briefly explaining.
Materials and methods
Lines 93-95. What does “tier five world class national teams” mean? Do you mean the top five teams in the world ranking?
Lines 96-97. 2-3 days per week is quite low for elite players. Are these training sessions with their national team, or does this include club training as well?
Lines 102-105. The FT3 group is significantly smaller (n=4) compared to FT1 (n=12) and FT2 (n=19). This is a major limitation that should be discussed.
Lines 112-113. The terminology is unclear. What does “48 individual observations” mean? Are these total data points from different matches?
Discussion
Lines 189-193. The lack of significant differences between FT2 and FT3 players is not explained adequately. This contrasts with male CP football studies, where FT3 players typically outperform FT2 players.
Lines 199-212. This section compares findings with male CP football but lacks a clear interpretation of why these differences exist. Could the smaller field size, lower intensity, or different tactical roles in female CP football explain these differences? The discussion should also highlight whether these differences are expected based on neuromuscular limitations in female CP athletes.
Lines 223-231. This is a crucial finding, but the explanation is too vague. Which factor is most likely responsible? If impairment was the key factor, then why do FT3 players not outperform FT2 players in high-speed running? The role of game structure (smaller field, fewer players) should be emphasized more clearly.
Lines 239-278. This paragraph is too descriptive. It states that acceleration and deceleration are important but does not analyze why female CP players perform differently than their male counterparts. Could muscle activation deficits, balance issues, or field size constraints explain the lower acceleration/deceleration loads?
Lines 294-316. The limitations section is well-written, but it should explicitly state how these limitations affected the results. The small sample size of FT3 players is a major issue, and its impact should be more clearly highlighted.
Author Response
Authors’ response (AR): We would like to thank the reviewer for the helpful advice and suggestions regarding this manuscript. In addition, we have found your criticism and recommendations very constructive. Also, we appreciate the work done in the revision of this manuscript, answering your concerns on a point-by-point basis. The changes to the paper have been made in red type in the new version of the manuscript.
- Introduction: The sentence is too long and difficult to follow. Suggest breaking it into two sentences for clarity.
AR: Thanks for the suggestion. To improve clarity and readability we have rewritten the sentence (Lines 39-45):
“Particularly, in the field of Para sports such as cerebral palsy (CP) football, the International Federation of Cerebral Palsy Football (IFCPF) has recently taken on the responsibility of promoting equality among para-athletes with eligible impairments of spastic hypertonia, ataxia, and dyskinesia [2]. This commitment aims to ensure inclusivity regardless of gender, ethnic origin, faith, or culture, focusing on footballers with motor problems due to CP or related underlying health conditions [2].
- The transition from discussing IPC’s mission to the focus on male CP football feels abrupt. Consider adding a bridging sentence explaining why this disparity exists.
AR: Thank you for this comment. We have added a sentence indicating the international development of CP Football; and indicated that, despite this, so far both tournaments and research have only focused on men generating a gap regarding the female modality. Hence, we have included the following piece of information (Lines 45-46):
“CP football is a widely developed team Para sport with both men's and women's divisions. However, almost all its international tournaments and scientific research have focused on the male modality.”
- The sentence is grammatically incorrect. “In the game. recent years” seems to be a typo. Suggest rewriting for clarity:“Nevertheless, female CP football has been growing exponentially as a competitive sport in recent years across all levels.”
AR: Thanks for this observation. We have written the suggested sentence (Lines 47-48):
“Nevertheless, female CP football has been growing exponentially as a competitive sport in recent years across all levels [3].”
- Good mention of differences between male and female CP football, but the connection to research gaps is weak. How do these differences impact performance, fatigue, or strategy? Add a sentence explaining why these factors warrant further study.
AR: Thank you for this comment. We have added the following sentence to improve the connection between ideas (Lines 63-65):
“In this sense, the external physical demands, as well as the fatigue and tactical behavior of both conventional and CP football players, vary depending on the individual playing space and playing time [10,12].”
- This sentence is poorly structured and difficult to understand.
AR: Thanks for this advice. We have rewritten it as follows (Lines 89-91):
“Given the significance of advancing scientific knowledge on competition in female CP football to align with the objectives established by the IPC and the IFCPF [1,21], researching this team Para sport modality is essential.”
- The objectives should explicitly state how they address the identified research gaps.
AR: Thank you for the comment. We have reduced the objectives to be more explicit and how address the identifies research gaps (Lines 91-94):
“Therefore, the main objectives of this study were (i) to describe the physical responses of female CP football players during their first international-level competition, and (ii) to analyze the differences in their physical responses based on the sport class of the players (i.e., FT1, FT2, and FT3).”
- The justification for this hypothesis is weak. Why is this expected? Are there physiological reasons or findings from male CP football that support this assumption? Consider briefly explaining.
AR: Thank you for this advice comment. We have reformulated the hypothesis since, considering studies in male CP football, we can only hypothesize that there will be differences depending on the sport class, because of the level of impairment of the players (Lines 94-98):
“Considering that differences in physical responses among male football players with CP based on sport class have been previously observed [9], we hypothesized that differences in physical responses will also exist among female football players with CP, because of the level impairment expected for each sport class.”
Materials and methods
- What does “tier five world class national teams” mean? Do you mean the top five teams in the world ranking?
AR: Thank you for this observation. The reviewer is correct. We have rewritten it to improve clarity (Lines 101-105):
“Thirty-five international-level [17] female CP football players (24.9 ± 8.3 yrs; 163.5 ± 7.0 cm; 59.0 ± 13.7 kg; 22.2 ± 3.5 kg·m-2) from top five national teams that competed in the International Federation of CP Football (IFCPF) 2022 World Cup (Salou, Spain), representing nearly (77.78%) all international players active in this team Para sport when the study took place.”
- Lines 96-97. 2-3 days per week is quite low for elite players. Are these training sessions with their national team, or does this include club training as well?
AR: Thank you for this comment. The training load information also includes club training practice. Although it is low for elite athletes, it can be explained by the fact that the female modality is an emerging modality and is in a development stage.
- The FT3 group is significantly smaller (n=4) compared to FT1 (n=12) and FT2 (n=19). This is a major limitation that should be discussed.
AR: Thank you for the comment. We have discussed it in the limitations section. While it is true that we have already indicated this in the limitations, we also add that the competition rules themselves only allow one FT3 player per team to be on the field. Therefore, although the analysis is carried out with four FT3 players, they represent a high percentage of the total number of FT3 players playing female CP Football (Lines 321-328):
“Particularly, the limited representation of players in the FT3 category constitutes a significant practical constraint that could have affected the obtained results, although competition rules limit each team to only having one FT3 player in the field of play [18]. Therefore, although the analysis is carried out with four FT3 players, they represent a high percentage of the total number of players in that sport class playing female CP football when the study was done. However, future research should aim to replicate the study with a larger sample as women's participation in international championships continues to increase.”
- The terminology is unclear. What does “48 individual observations” mean? Are these total data points from different matches?
AR: Each individual observation means the data of the physical responses monitored in a single game by one player. This clarification has been included in the Procedures subsection (Lines 120-122):
“Physical responses were recorded during official matches of the international competition (n = 35 players, 4 official matches, and 48 individual observations −i.e., physical responses monitored in a single game by one player−, range = 1-3 matches per player).”
Discussion
- The lack of significant differences between FT2 and FT3 players is not explained adequately. This contrasts with male CP football studies, where FT3 players typically outperform FT2 players.
AR: Thank you for the comment. In these lines our intention is to recall the main results, in order to address the concrete explanations in the following paragraphs. However, to expand the explanations regarding the limited differences found between these profiles, authors have included the following piece of information (Lines 302-305):
“The limited significant differences between FT2 and FT3 players could be attributed to their higher functional ability, combined with the smaller pitch size and the specific constraints of the modality (e.g., five-a-side game), which may limit variations in time-motion variables.”
- This section compares findings with male CP football but lacks a clear interpretation of why these differences exist. Could the smaller field size, lower intensity, or different tactical roles in female CP football explain these differences? The discussion should also highlight whether these differences are expected based on neuromuscular limitations in female CP athletes.
AR: Thanks for this observation. We have added the following sentence (Lines 227-229):
“A smaller field size and the smaller number of players could explain the difference in results compared to the results in men, as female CP football players may adopt different tactical roles.”
- Lines 223-231. This is a crucial finding, but the explanation is too vague. Which factor is most likely responsible? If impairment was the key factor, then why do FT3 players not outperform FT2 players in high-speed running? The role of game structure (smaller field, fewer players) should be emphasized more clearly.
AR: Thank you for this advice. We have added the following explanation based on the game format (Lines 229-234):
“In this regard, in the male format (7-a-side, field dimensions: 70×50 m), the individual interaction space is 250 m², whereas in the female CP football format (5-a-side, field dimensions: 40×27 m), the individual interaction space is 108 m². The fact that the individual interaction space is significantly smaller could have influenced the ability of female players to reach their maximum physical potential in distances covered at medium-to-high intensities.”
- Lines 239-278. This paragraph is too descriptive. It states that acceleration and deceleration are important but does not analyze why female CP players perform differently than their male counterparts. Could muscle activation deficits, balance issues, or field size constraints explain the lower acceleration/deceleration loads?
AR: Thank you for the comment. We understand the reviewer’s concern. However, in the present study, we have focused on discussing the differences or lack thereof in accelerations and decelerations among players of the different CP football sport classes, in line with our objectives. We acknowledge that the comparison between male and female players is relevant, but this comparison falls outside the scope of this study. Nonetheless, we appreciate the reviewer’s valuable input and will consider analyzing and discussing this aspect in a future study, providing data on both male and female players in a similar game context.
- The limitations section is well-written, but it should explicitly state how these limitations affected the results. The small sample size of FT3 players is a major issue, and its impact should be more clearly highlighted.
AR: Thanks for the advice. While it is true that we have already indicated this in the limitations, we also add that the competition rules themselves only allow one FT3 player per team to be on the field. Therefore, although the analysis is carried out with four FT3 players, they represent a high percentage of the total number of FT3 players playing female CP football (Lines 321-328):
“Particularly, the limited representation of players in the FT3 category constitutes a significant practical constraint that could have affected the obtained results, although competition rules limit each team to only having one FT3 player in the field of play [18]. Therefore, although the analysis is carried out with four FT3 players, they represent a high percentage of the total number of players in that sport class playing female CP football when the study was done. However, future research should aim to replicate the study with a larger sample as women's participation in international championships continues to increase.”
Round 2
Reviewer 1 Report
Comments and Suggestions for Authors
It is ok now
Reviewer 3 Report
Comments and Suggestions for Authors
The authors have addressed all the comments as requested.